# Therapeutic Target Discovery for Multiple Myeloma: Identifying Druggable Genes via Mendelian Randomization

**DOI:** 10.3390/biomedicines13040885

**Published:** 2025-04-05

**Authors:** Shijun Jiang, Fengjuan Fan, Qun Li, Liping Zuo, Aoshuang Xu, Chunyan Sun

**Affiliations:** Institute of Hematology, Union Hospital, Tongji Medical College, Huazhong University of Science and Technology, Wuhan 430022, China; guazhoudu123@sina.com (S.J.); kittyfan0120@163.com (F.F.); liqun97@126.com (Q.L.); zuo5258@126.com (L.Z.); xas_medicine@163.com (A.X.)

**Keywords:** multiple myeloma, mendelian randomization, *ORM1*, *OVGP1*, pregnanolone, irinotecan

## Abstract

**Background**: Multiple myeloma (MM) is a hematological malignancy originating from the plasma cells present in the bone marrow. Despite significant therapeutic advancements, relapse and drug resistance remain major clinical challenges, highlighting the urgent need for novel therapeutic targets. **Methods**: To identify potential druggable genes associated with MM, we performed Mendelian randomization (MR) analysis. Causal candidates were further validated using a single-tissue transcriptome-wide association study (TWAS), and colocalization analysis was conducted to assess shared genetic signals between gene expression and disease risk. Potential off-target effects were assessed through an MR phenome-wide association study (MR-PheWAS). Additionally, molecular docking and functional assays were used to evaluate candidate drug efficacy. **Results**: The MR analysis identified nine druggable genes (FDR < 0.05), among which Orosomucoid 1 (*ORM1*) and Oviductal Glycoprotein 1 (*OVGP1*) were supported by both TWAS and colocalization evidence (PPH4 > 0.75). Experimental validation demonstrated the significant downregulation of ORM1 and OVGP1 in MM cells (*p* < 0.05). Pregnenolone and irinotecan, identified as agonists of ORM1 and OVGP1, respectively, significantly inhibited MM cell viability, while upregulating their expression (*p* < 0.05). **Conclusions**: Our study highlights ORM1 and OVGP1 as novel therapeutic targets for MM. The efficacy of pregnenolone and irinotecan in suppressing MM cell growth suggests their potential for clinical application. These findings provide insights into MM pathogenesis and offer a promising strategy for overcoming drug resistance.

## 1. Introduction

Multiple myeloma (MM) is a blood cancer characterized by the unchecked expansion of plasma cells in the bone marrow. Plasma cells, a subset of B lymphocytes responsible for antibody production, undergo malignant transformation in MM, leading to uncontrolled proliferation and the secretion of abnormal monoclonal immunoglobulins (M-proteins). This aberrant cell growth disrupts normal hematopoiesis, causing clinical symptoms such as anemia, recurrent infections, bone fragility, fractures, and renal dysfunction. MM accounts for approximately 1% of all cancers and approximately 10% of blood cancers globally. Recent estimates indicate that the global incidence of MM is 2.1 per 100,000 individuals annually, with a mortality rate of 1.2 per 100,000 [1]. The condition predominantly impacts elderly individuals, with a median age of diagnosis of 69 years, and is slightly more common in males.

The etiology of MM remains incompletely understood, but accumulating evidence suggests that genetic predisposition, chronic inflammation, and environmental exposures are major contributing factors [2,3,4]. Genome-wide association studies (GWASs) have identified multiple susceptibility loci associated with MM risk, including variants of TULK4, ATG5, and WAC [5]. Notably, mutations in KRAS, NRAS, TP53, and MYC are frequently observed in MM, playing key roles in disease progression [6]. Additionally, chromosomal translocations involving the IGH loci—such as t(4;14), t(11;14), and t(14;16)—have been implicated in MM pathogenesis [7]. The bone marrow microenvironment is crucial in MM progression, providing survival signals through cytokines (e.g., IL-6, TNF-α, TGF-β, etc.), adhesion molecules, and extracellular matrix interactions [8,9,10]. MM cells frequently exploit these interactions to evade immune surveillance and resist apoptosis [11]. The dysregulation of several key signaling pathways, including RAS/MAPK, NF-κB, JAK/STAT, and PI3K/AKT/mTOR, further contributes to tumor growth and drug resistance [12].

Despite significant therapeutic advances, MM remains an incurable disease. The introduction of proteasome inhibitors (e.g., bortezomib, carfilzomib), immunomodulatory drugs (e.g., lenalidomide, pomalidomide), and monoclonal antibodies (e.g., daratumumab, elotuzumab) has substantially improved patient outcomes [13]. However, most patients eventually experience relapse or develop resistance to treatment, necessitating the development of novel therapeutic strategies [14]. Several mechanisms contribute to drug resistance in MM, including clonal evolution, epigenetic modifications, and alterations in drug metabolism pathways [15,16]. For instance, mutations in proteasome subunits (PSMB5) reduce bortezomib sensitivity, while the upregulation of anti-apoptotic proteins (BCL2, MCL1, etc.) enhances survival signaling [17,18]. The recent emergence of chimeric antigen receptor (CAR)-T cell therapy, bispecific T cell engagers (BiTEs), and antibody–drug conjugates (ADCs) has provided promising alternatives [19]. However, these therapies are expensive and are associated with significant immune-related toxicities, highlighting the need for novel, cost-effective druggable targets.

Targeting druggable genes is a promising strategy for overcoming drug resistance and improving treatment outcomes [20]. For example, B Cell Lymphoma 2 (BCL2) inhibitors such as venetoclax have shown efficacy in patients with t(11;14) translocations, providing a personalized treatment approach. Additionally, inhibitors targeting the MEK/ERK and JAK/STAT pathways have demonstrated potential in preclinical and early clinical studies, underscoring their relevance to the pathophysiology of MM.

Mendelian randomization (MR) has emerged as a powerful approach for investigating causal relationships between genetic variants and disease outcomes. This method utilizes genetic variants, such as instrumental variables (IVs), to infer causality, while minimizing the impact of confounding factors and reverse causation, thereby mimicking a randomized controlled trial (RCT) at the genetic level [21]. Since genetic variants are randomly assigned at conception according to Mendel’s laws, their associations with disease risk factors are generally less influenced by environmental or behavioral factors. MR has been widely applied in epidemiological and genetic studies to uncover the causal links between risk factors and disease outcomes. For example, MR has provided evidence that high-density lipoprotein cholesterol (HDL-C) is not causally protective against coronary artery disease, challenging prior observational findings [22]. Furthermore, MR has been instrumental in evaluating the causal effects of lifestyle factors, such as smoking and alcohol consumption, on various health outcomes, thereby informing public health policies and intervention strategies [23].

A specialized application of MR, drug target Mendelian randomization (DTMR), integrates genome-wide association study (GWAS) data with genetic variants located in druggable genes to infer potential therapeutic targets. DTMR is particularly valuable in oncology, where it enables the identification of genes that not only contribute to disease risk, but also represent viable intervention points for drug development [24,25]. For instance, Wang et al. employed a proteome-wide MR analysis to integrate plasma proteomic data with GWAS data, identifying 13 proteins causally associated with MM risk, including nicotinamide phosphoribosyl transferase (NAMPT) and suppressor of cytokine signaling 3 (SOCS3) [26]. This approach has been applied to study key regulatory pathways in diseases, including bone marrow microenvironment interactions, immune modulation, and cell signaling networks, thereby accelerating drug discovery and facilitating clinical translation [27,28]. Given the complex pathophysiology of MM, which involves dynamic interactions between malignant plasma cells and the bone marrow microenvironment, DTMR provides a robust strategy to prioritize genes for targeted therapy.

In this study, we employed MR-based approaches to identify and validate novel druggable targets for MM. By integrating GWAS data, colocalization analysis, and experimental validation, we aimed to identify new druggable genes causally associated with MM risk, validate these targets through colocalization analysis and in vitro experiments, assess potential off-target effects using an MR phenome-wide association study (MR-PheWAS), and evaluate the therapeutic potential of candidate drugs through molecular docking and functional assays. Our findings provide new insights into MM pathogenesis and may facilitate the development of targeted therapies aimed at overcoming drug resistance and improving patient outcomes.

## 2. Materials and Methods

### 2.1. Study Design

The STROBE-MR checklist was completed to ensure the integrity of this observational MR study (Appendix A) [29]. This study integrated both previously established methodologies and newly implemented approaches to investigate the causal relationship between druggable gene expression and MM susceptibility and progression. The workflow consisted of five key steps:

Two-sample Mendelian randomization (MR) was used to assess causality between cis-eQTL-associated druggable genes and MM susceptibility. This method has been widely used in genetic epidemiology, and follows established procedures [30].

To further assess whether these genes contribute to MM susceptibility via gene expression regulation, we performed a single-tissue transcriptome-wide association study (TWAS). A TWAS investigates the impact of gene expression on disease risk in a tissue-specific manner by utilizing expression prediction models generated from large-scale transcriptomic resources such as GTEx, eQTLGen, and DGN [31,32].

Bayesian colocalization analysis was conducted to confirm whether shared genetic variants influence both gene expression and MM risk. This study followed standard colocalization approaches, but the choice of PPH4 > 0.75 as a threshold was specifically optimized for MM [33].

A Mendelian randomization phenome-wide association study (MR-PheWAS) was employed to systematically explore the potential off-target effects of ORM1 and OVGP1 modulation (a novel addition to this research pipeline to evaluate safety risks across multiple traits).

In silico molecular docking analysis and in vitro validation:

Computational molecular docking was used to predict candidate small-molecule agonists for ORM1 and OVGP1, a method commonly used in drug discovery. However, the selection of compounds from DSigDB, rather than a general compound library, is a novel approach tailored to MM drug repurposing [34,35].

In vitro functional assays (qRT-PCR, Western blot, CCK-8, etc.) were performed to validate the efficacy of the identified small-molecule agonists, following standardized experimental procedures, but extending their application to ORM1 and OVGP1 in MM cells.

This integrative framework combines existing methodologies with novel analytical pipelines, particularly the combined use of MR-PheWAS and molecular docking in MM drug target discovery, thus advancing the field of genetics-driven target identification (Figure 1).

### 2.2. Antibodies and Reagents

The marker used was the WJ103 model from Yamei, China. The antibodies used for Western blotting were ORM1 (16439-1-AP, Proteintech, Wuhan, China, 1:1000), MATN2 (24064-1-AP, Proteintech, Wuhan, China, 1:1000), OVGP1 (22324-1-AP, Proteintech, China, 1:1000), and GAPDH (10494-1-AP, Proteintech, Wuhan, China, 1:10,000). The secondary antibody used was horseradish peroxidase (HRP)-conjugated goat anti-rabbit antibody (ANT020; Antgen, Wuhan, China; 1:10,000). All the antibodies were diluted in Tris-buffered saline with Tween, supplemented with either 5% skim milk or bovine serum albumin, according to the manufacturer’s instructions.

Pregnanolone (MedChemExpress, catalog: HY-B0151, Shanghai, China) and irinotecan (MedChemExpress, catalog: HY-16562, Shanghai, China) were used. All drugs were dissolved in dimethyl sulfoxide (DMSO) and stored at −20 °C in the dark.

The primers for the real-time qPCR targeting of ORM1, MATN2, OVGP1, and GAPDH were purchased from Beijing Qingke Biotechnology Co., Ltd., Beijing China. The primer sequences are provided in Table 1.

### 2.3. Exposure Data

Finan et al. identified 4479 druggable genes in total, which were categorized into three tiers according to their relevance in therapeutic development. Tier 1, which includes 1427 genes, consists of targets for licensed small molecules and biological therapies, as well as those currently in clinical development. Tier 2, consisting of 682 genes, represents those encoding proteins that bind to bioactive small molecules or share substantial sequence similarity with existing drug targets. Tier 3, with 2370 genes, encompasses genes responsible for extracellular or secreted proteins of major druggable families, such as G-protein-coupled receptors (GPCRs), ion channels, kinases, and nuclear hormone receptors, which are not part of the first two tiers [36]. Among these genes, 2902 are linked to the cis-eQTL data from the eQTLGen Consortium, which integrates genomic information from 37 datasets encompassing 31,684 individuals of European descent. These cis-eQTL data underscore the connection between gene expression levels and genetic variants within a 1 Mb region surrounding the gene, with a focus on variants having a minor allele frequency (MAF) greater than 0.01 [37]. These findings provide a comprehensive framework for understanding the therapeutic potential of druggable genes (Appendix A).

### 2.4. Instrumental Variable (IV) Selection

The cis-eQTLs associated with potential druggable genes from the eQTLGen database were selected as IVs for MR analysis [38,39]. To guarantee the reliability and robustness of these IVs, a set of rigorous criteria was applied:

#### 2.4.1. Statistical Significance (*p* < 5 × 10^−8^)

A genome-wide significance threshold was set at *p* < 5 × 10^−8^, which is commonly used in GWAS and eQTL studies to minimize false-positive associations [40]. This threshold ensures that only high-confidence eQTLs are included, reducing the influence of weak genetic instruments.

#### 2.4.2. Linkage Disequilibrium (LD) Clumping (r^2^ < 0.1; Window Size = 10,000 kb)

To maintain independence among IVs, we applied LD clumping using a reference panel from the 1000 Genomes European population. An r^2^ < 0.1 threshold ensured that the selected SNPs were not in high LD, minimizing collinearity and bias in MR estimates. The 10,000 kb window was chosen to prioritize variants with the strongest cis-regulatory effects on the target gene, while avoiding distant trans-eQTLs.

#### 2.4.3. Instrument Strength (F-Statistic > 10)

The F-statistic was used to quantify instrument strength, ensuring that weak instruments did not bias MR estimates. According to Staiger and Stock’s rule of thumb, an F-statistic > 10 is necessary to minimize weak instrument bias in MR studies [41]. The F-statistic is computed using the following formula:$$F=\frac{R^{2}}{\left(1-R^{2}\right)}\times \frac{\left(N-k-1\right)}{k}.$$ where R^2^ represents the proportion of variance explained by the IV, N is the sample size, and k is the number of instruments.

#### 2.4.4. Software and Filtering Parameters

PLINK (version 1.9) was used for LD pruning and variant filtering.

By applying the aforementioned criteria, a high-quality set of IVs was selected for MR analysis to investigate the causal link between druggable gene expression and the risk of MM.

### 2.5. Outcome Data

The outcome data were derived from the GWAS dataset of MM in the UKB Scalable and Accurate Implementation of Generalized Mixed Model (SAIGE) database for association analysis. The UKB-SAIGE database, one of the largest health and genetic databases in the world, includes detailed health and genetic information of approximately 500,000 participants in the UK, facilitating extensive genetic research on complex diseases [42].

In this study, data from 552 patients with MM and 404,466 controls were utilized. The SAIGE model efficiently handles large-scale data, also accounting for genetic correlations, sex, birth year, and principal components for population stratification [43]. This model is particularly effective in the case of rare diseases, reducing false positives and improving the detection of rare variants.

### 2.6. Two-Sample Mendelian Randomization Analysis (Two-Sample MR)

MR analysis was conducted via the R package “TwoSampleMR” (version 0.5.6). For analyses involving a Single-Nucleotide Polymorphism (SNP) as the IV, the Wald ratio method was used to estimate the causal effects. For analyses involving multiple SNP IVs, five established techniques were utilized, i.e., inverse variance weighted (IVW), MR-Egger, weighted median, simple mode, and weighted mode [44]. Among these methods, the IVW approach was selected as the primary analytical tool because of its high efficiency and reliability, assuming that all SNPs function as valid IVs without horizontal pleiotropy. By integrating SNP-specific causal estimates through inverse variance weighting, the IVW method optimizes statistical power while minimizing standard errors, making it the most commonly employed technique in MR studies when the validity of instruments is ensured.

FDR corrections were performed to control for multiple testing, thereby enhancing the dependability of the significant findings. The final results were primarily based on the estimates derived from IVW, given its superior statistical robustness and precision, assuming valid instruments.

Sensitivity analyses:

Heterogeneity test (Cochran’s Q-statistic): Cochran’s Q test was used to assess heterogeneity among the IVs. A significant Q-statistic (*p* < 0.05) indicates potential heterogeneity in causal effects, necessitating additional tests, such as MR–Egger regression.

Directional pleiotropy test (MR–Egger intercept): MR–Egger regression was performed to evaluate the presence of directional pleiotropy, which occurs when IVs influence the outcome through pathways other than the exposure gene. A significant MR–Egger intercept (*p* < 0.05) suggests the presence of pleiotropy, in which case, weighted median MR is considered [45].

This comprehensive methodology enabled a robust causal inference for exposures and outcomes, while addressing biases and confounding factors.

### 2.7. Transcriptome-Wide Association Study (TWAS)

To further validate candidate causal genes for MM identified through two-sample Mendelian randomization (MR), we conducted a single-tissue TWAS using the FUSION pipeline [46]. This approach integrates GWAS summary statistics with gene expression prediction models derived from GTEx v8 whole blood data from individuals of European ancestry (GTExv8.EUR.Whole_Blood).

Cis-regulated expression was imputed within a ±500 kb window around each gene using pre-trained expression weights. To accurately model linkage disequilibrium (LD) among SNPs in the cis region, we employed the 1000 Genomes Project Phase 3 European LD reference panel provided by FUSION in PLINK binary format, stratified by chromosome.

The TWAS analysis was limited to genes prioritized by two-sample MR, and statistical significance was evaluated using false discovery rate (FDR) correction. Genes with FDR-adjusted *p*-values < 0.05 were considered significantly associated with MM susceptibility.

### 2.8. Colocalization

The false positives resulting from LD were eliminated through the Bayesian colocalization analysis. This method evaluates whether exposure (e.g., gene expression) and outcomes (e.g., disease) are driven by the same genetic variant. This colocalization analysis led to five hypotheses:

**H0.** *Neither gene expression nor disease is associated with causal variants in the region*.

**H1.** *Only gene expression is associated with a causal variant*.

**H2.** *Only the disease is associated with a causal variant*.

**H3.** *Gene expression and disease are linked to separate causal variants*.

**H4.** *Gene expression and disease share a common causal variant*.

#### 2.8.1. Single Causal Variant per Region

For this analysis, we used the “coloc” R package (version 4.0.6) [47]. The “coloc” model assumes that each genomic locus contains at most one causal variant affecting either the gene expression or disease risk. If multiple independent variants exist within the same region, this assumption may be violated, leading to misinterpretation of colocalization results.

#### 2.8.2. Effect Sizes Are Derived from GWAS and eQTL Summary Statistics

The method assumes that effect sizes and standard errors are correctly estimated from large-scale GWAS and eQTL studies. Poorly estimated effect sizes can lead to incorrect posterior probabilities.

#### 2.8.3. Rationale for PPH_4_ > 0.75 Threshold

We adopted PPH_4_ > 0.75 as the colocalization threshold based on the previous literature and statistical considerations: a threshold of 0.75–0.80 is widely used in genetic studies to define high-confidence colocalization [48,49]. PPH_4_ values close to 1.0 indicate strong colocalization, while values below 0.50 suggest weak or no colocalization. This threshold balances specificity and sensitivity, ensuring that selected genes are likely causal, while minimizing false positives.

#### 2.8.4. Visualization of Colocalization Results

Colocalization results were visualized using the “locuscomparer” R package (version 1.0.0), which generates regional association plots comparing GWAS and eQTL signals within the selected loci. These visualizations help to confirm whether the eQTL and GWAS peaks align, supporting shared genetic regulation.

### 2.9. Mendelian Randomization Phenome-Wide Association Study (MR-PheWAS)

An MR-PheWAS was utilized to investigate whether these genes might serve as potential therapeutic targets for various diseases. This study specifically employed cis-eQTLs of two core proteins, ORM1 and OVGP1, as exposure variables, while analyzing 1403 disease phenotypes from the UKB-SAIGE dataset as outcomes. By systematically conducting thousands of MR analyses, this approach examined the causal influence of genetic variants across a broad spectrum of disease traits. The IVW method was employed to estimate causal relationships, with statistical significance defined as an FDR of <0.05. This robust analytical framework provides a comprehensive means to uncover novel associations between gene expression and disease phenotypes, offering critical insights into potential therapeutic targets [50,51].

This comprehensive approach allows for the identification of novel associations between gene expression and disease phenotypes, offering critical insights into potential therapeutic targets. For instance, a recent MR-PheWAS investigating the causal role of immune response to varicella zoster virus (VZV) in multiple traits identified associations with conditions such as Dupuytren disease, mononeuropathies of the upper limb, sarcoidosis, coeliac disease, and teeth problems. This study highlighted the potential of MR-PheWASs to uncover previously unrecognized links between immune response and various health conditions [52].

Furthermore, another MR-PheWAS focusing on insomnia symptoms revealed potential causal effects on a diverse range of outcomes, including anxiety, depression, pain, body composition, respiratory, musculoskeletal traits, and cardiovascular traits. This underscores the broad applicability of MR-PheWASs in elucidating the impact of specific exposures on multiple health outcomes [53].

By systematically conducting thousands of MR analyses, an MR-PheWAS serves as a powerful tool to uncover novel associations between gene expression and disease phenotypes, thereby offering critical insights into potential therapeutic targets.

### 2.10. Molecular Docking

To investigate the potential interactions between OVGP1 and ORM1 and their respective agonists, we performed molecular docking simulations using AutoDock Vina. This computational approach enabled us to predict the binding affinity and interaction modes of selected ligand molecules with their target proteins.

#### 2.10.1. Retrieval and Preparation of Protein Structures

The three-dimensional structures of OVGP1 and ORM1 were obtained from the UniProt database: for ORM1, the crystal structure was obtained from 3KQ0 (PDB ID), and for OVGP1, the predicted structure was obtained from AF-Q12889-F1 (AlphaFold ID). Protein structures were preprocessed using AutoDockTools (version 1.5.7), including the removal of water molecules, the addition of polar hydrogens to account for hydrogen bonding interactions, and the assignment of Gasteiger charges to ensure accurate electrostatic calculations.

#### 2.10.2. Selection and Preparation of Ligands

The potential agonists for OVGP1 and ORM1 were identified through the Drug Signatures Database (DSigDB), prioritizing compounds with reported gene expression modulation effects. Ligand structures were downloaded from PubChem, and optimized using Open Babel (version 3.1.1) to convert them into an AutoDock-compatible PDBQT format [54].

#### 2.10.3. Docking Parameters and Scoring Functions

Molecular docking was conducted using AutoDock Vina (version 1.2.3) [55], employing the following grid box settings: ORM1: center at x = 32.1, y = 27.8, z = 40.7, with grid dimensions of 60 × 60 × 60 Å; OVGP1: center at x = 45.2, y = 38.6, z = 52.4, with grid dimensions of 60 × 60 × 60 Å. A grid spacing of 1.0 Å was used to ensure there was sufficient search space for flexible docking. The AutoDock Vina scoring function was used to rank docking poses based on their binding energy (ΔG, kcal/mol), where lower ΔG values indicate a higher ligand–protein affinity [56].

#### 2.10.4. Visualization and Interaction Analysis

The top-ranked docking complexes were visualized using PyMOL (version 2.5.5) to analyze binding site interactions.

The results provided intuitive insights and reliably supported the interaction between potential ligands and target proteins.

### 2.11. Cell Lines and Cell Culture

Human MM cell lines (RPMI-8226, NCI-H929, MM.1S, and OPM-2) were acquired from the Cell Bank of the Chinese Academy of Sciences, Beijing, China. GM12878 cells were purchased from Shanghai QuiCell Biotechnology Co., Ltd., Shanghai, China. The cells were cultured in RPMI-1640 medium enriched with 10% FBS and 1% penicillin-streptomycin, and incubated at 37 °C under a 5% CO_2_ atmosphere with 95% humidity. They were passaged when their density reached 1 ×10^6^–2 × 10^6^ cells/mL at a 1:2 or 1:3 ratio, ensuring exponential growth.

### 2.12. RT-qPCR

To ensure the accurate quantification of gene expression, we performed reverse transcription quantitative PCR (RT-qPCR), following the Minimum Information for Publication of Quantitative Real-Time PCR Experiments (MIQE) guidelines [57].

#### 2.12.1. RNA Extraction and Quality Control

Total RNA was extracted from cells using the TRIzol reagent (Invitrogen, Waltham, MA, USA) and by following the manufacturer’s protocol. RNA purity and concentration were assessed using a NanoDrop 2000 spectrophotometer (Thermo Fisher Scientific, Waltham, MA, USA) by measuring the A260/A280 and A260/A230 ratios. Samples with an A260/A280 ratio between 1.8 and 2.0 were considered highly pure.

#### 2.12.2. cDNA Synthesis

Reverse transcription (RT) was performed using the HiScript III RT SuperMix Kit (Vazyme, Nanjing, China). Genomic DNA contamination was eliminated by including a gDNA wiper step before cDNA synthesis. cDNA synthesis reactions were carried out in a total volume of 20 µL, with 500 ng of total RNA per reaction.

#### 2.12.3. RT-qPCR Experimental Conditions

qPCR was conducted using the QuantStudio 5 Real-Time PCR System (Applied Biosystems, Waltham, MA, USA) in a 20 µL reaction volume containing 10 µL ChamQ SYBR qPCR Master Mix (Vazyme, Nanjing, China), 0.4 µL forward primer (10 µM), 0.4 µL reverse primer (10 µM), 1 µL cDNA template, and 8.2 µL RNase-free water. Melt curve analysis was performed from 65 °C to 95 °C to confirm reaction specificity.

#### 2.12.4. Normalization and Quantification Method

The ΔΔCt method was used for relative gene expression analysis [58]. GAPDH was selected as the reference gene, after confirming its stable expression across experimental conditions.

### 2.13. Western Blotting

Western blotting was performed to evaluate the expression of target proteins [59].

#### 2.13.1. Protein Extraction and Quantification

Cell lysates were prepared using RIPA lysis buffer (Beyotime, Shanghai, China) supplemented with protease and phosphatase inhibitors (Thermo Fisher, Waltham, MA, USA) to prevent protein degradation. Samples were incubated on ice for 30 min, with vortexing carried out every 5 min, followed by centrifugation at 12,000× *g* for 15 min at 4 °C, using an Eppendorf 5430 R centrifuge (Hamburg, Germany), to remove cell debris. Protein concentration was measured using the Bicinchoninic Acid (BCA) Assay with a NanoDrop 2000 spectrophotometer (Thermo Fisher, Waltham, MA, USA). Loading amounts were normalized to ensure equal protein input across all samples.

#### 2.13.2. SDS-PAGE and Membrane Transfer

Equal protein quantities (20–30 µg per lane) were loaded and separated via sodium dodecyl sulfate–polyacrylamide gel electrophoresis (SDS-PAGE, 12% gels) using a Bio-Rad Mini-PROTEAN Tetra System (Hercules, CA, USA). Proteins were transferred to PVDF membranes (Millipore, Burlington, MA, USA) using a Bio-Rad Trans-Blot Cell wet transfer system at 100 V for 1.5 h at 4 °C.

#### 2.13.3. Blocking and Antibody Incubation

Blocking step: membranes were incubated with 5% skim milk (Bio-Rad, Hercules, CA, USA) in Tris-buffered saline with 0.1% Tween-20 (TBST) for 1 h at room temperature, to reduce nonspecific binding. Primary antibody incubation: membranes were incubated with primary antibodies overnight at 4 °C, with gentle shaking. Secondary antibody incubation: after three washes (5 min each) with TBST, membranes were incubated with secondary antibodies for 1 h at room temperature. Protein detection: bands were visualized using an Enhanced Chemiluminescence (ECL) (Beyotime, Shanghai, China) substrate and detected using a GE Healthcare Amersham Imager 600 (Chicago, IL, USA).

#### 2.13.4. Quantification of Band Intensity

Band intensity was quantified using the ImageJ software (version 1.54d) (Bethesda, MD, USA), and protein expression levels were normalized with the loading control (GAPDH).

### 2.14. Cell Counting Kit-8 (CCK-8) Assays

To evaluate the proliferation and viability of multiple myeloma (MM) cells in response to drug treatment, we performed CCK-8 assays, following standardized in vitro validation protocols.

#### 2.14.1. Experimental Conditions

Used cell lines: RPMI-8226 and NCI-H929. Seeding density: 1 × 10⁴ cells per well in 96-well plates (Corning, Corning, NY, USA).

#### 2.14.2. Drug Treatment and Experimental Setup

Tested compounds: pregnenolone (ORM1 agonist) and irinotecan (OVGP1 agonist). Concentration gradient: cells were treated with 10, 20, 30, and 40 μM of each compound. Control conditions: DMSO (0.1%) was used as a negative control (vehicle control) to rule out solvent effects. Untreated cells served as an additional control to measure baseline proliferation.

#### 2.14.3. CCK-8 Assay

Drug incubation: after plating, cells were treated with the compounds for 24 h at 37 °C in 5% CO_2_. Addition of CCK-8 reagent: a 10 μL volume of CCK-8 solution (Beyotime, China) was added to each well. Incubation: cells were incubated for 1 h at 37 °C, allowing viable cells to reduce the CCK-8 substrate into a formazan product. Absorbance measurement: the optical density (OD) at 450 nm was measured using a microplate reader (Thermo Fisher Scientific, Waltham, MA, USA).

#### 2.14.4. Data Analysis and Statistical Methods

Normalization: The OD values were normalized with the negative control (DMSO-treated group) to calculate relative viability. Background absorbance (from only the medium) was subtracted from all readings.

Calculation of cell viability (%):(1)$$\mathrm{Cell}\;\mathrm{viability}=\left(\frac{{\mathrm{OD}}_{\mathrm{treated}}-{\mathrm{OD}}_{\mathrm{blank}}}{{\mathrm{OD}}_{\mathrm{control}}-{\mathrm{OD}}_{\mathrm{blank}}}\right)\times 100.$$

IC50 determination: the half-maximal inhibitory concentration (IC50) was calculated using nonlinear regression curve fitting (log [inhibitor] vs. response, variable slope model) in GraphPad Prism (version 9.0).

### 2.15. Statistical Analysis

All quantitative experiments were performed in triplicate or higher, unless otherwise specified, and values are reported as mean ± standard deviation (SD). Statistical analyses were carried out using GraphPad Prism 9 (GraphPad Software, La Jolla, CA, USA). For comparisons between two groups, an unpaired two-tailed Student’s *t*-test was used. For multi-group comparisons (e.g., multiple cell lines or drug concentrations), MR and TWAS analyses were conducted as described, with multiple-testing corrections applied for high-dimensional analyses (the FDR method). Significance was defined as a *p* of <0.05 (or an FDR of <0.05 for adjusted results).

## 3. Results

### 3.1. Identification of Exposure Genes

On the basis of the selection criteria (*p* < 5 × 10^−8^, r^2^ < 0.1, window size of 10,000 kb, and F statistic > 10), 2533 potential druggable genes were identified. These genes were selected to ensure their relevance to MM and statistical significance through rigorous bioinformatic and genetic screening processes.

A total of 40,447 SNPs were identified as IVs for these 2533 druggable genes, meeting the specified criteria. All IVs had an F-statistic of >10, suggesting the absence of any weak instrumental bias. The IVs are explained in detail in Appendix A.

### 3.2. Two-Sample MR Analysis Validated Druggable Genes

After identifying the 2533 potential druggable genes, two-sample MR analysis was used to evaluate the causal link between MM risk and the expression of these genes. To improve accuracy, all *p*-values were examined via FDR correction. After correction, 34 genes were significantly associated with MM risk (FDR < 0.05) (Figure 2).

Heterogeneity test: The reliability of the results was ensured by performing heterogeneity tests on the 34 significant genes. These tests examined the variability of effects across different IVs to confirm consistency in the causal link between gene expression and the risk for MM. The findings indicated consistent association effects across IVs, and no significant heterogeneity was observed (Appendix A).

Horizontal Pleiotropy Test: Horizontal pleiotropy tests were performed to exclude the possibility that the IVs influence MM risk through pathways other than target gene expression. Using methods such as MR-Egger regression, the evidence of horizontal pleiotropy was noted only for Carbonic Anhydrase 2 *(CA2)* and Tumor Necrosis Factor *(TNF)* (*p* < 0.05), further supporting the causal role of these genes in MM (Appendix A).

### 3.3. Transcriptome-Wide Association Study (TWAS) Identified Susceptibility Genes for MM

We performed a single-tissue TWAS using gene expression prediction models from GTExv8.EUR.Whole_Blood to evaluate associations between genetically predicted expression and multiple myeloma (MM) risk for candidate genes identified by two-sample Mendelian randomization. The analysis identified four genes that remained statistically significant after false discovery rate (FDR) correction (FDR < 0.05): *OVGP1*, *ORM1*, *ALOX5AP*, and *PTGDS* (Table 2).

Among them, *OVGP1* (rs1264878) exhibited the strongest negative association with MM risk (Z = −3.4), suggesting that increased genetically predicted expression of *OVGP1* in whole blood may be associated with a reduced risk of MM. Similarly, *ORM1* (rs7851482) and *ALOX5AP* (rs6490461) also showed negative TWAS Z-scores (Z = −2.95 and −2.8571), indicating potential protective roles. In contrast, *PTGDS* (rs2811786) demonstrated a positive association (Z = 2.75), implying that higher expression of *PTGDS* may contribute to increased MM susceptibility.

These findings provide transcriptome-level evidence supporting the involvement of these genes in MM pathogenesis, and warrant further investigation to elucidate their functional roles in disease development.

### 3.4. Colocalization Confirmed Shared Genetic Variants for Gene Expression and MM

Colocalization analysis was conducted to verify whether a single genetic variant contributes to both gene expression and disease phenotypes. This analysis calculated the posterior probability of shared causal variants (PPH4) to determine shared causality (Appendix A). Among the identified genes, only Orosomucoid 1 *(ORM1)*, Matrilin-2 *(MATN2)*, and Oviductal Glycoprotein 1 *(OVGP1)* had PPH4 values > 0.75, strongly indicating that these genes are causally linked to MM through the same genetic variants (Figure 3).

The multi-step pipeline applied in this study—including exposure gene identification, two-sample MR, TWAS, colocalization, and functional validation—represents a systematic framework for identifying and prioritizing therapeutic targets in MM. While each individual method has been previously used in complex disease genetics, the comprehensive integration of these approaches in MM research is a novel aspect of this study.

Similar strategies have been employed to identify causal genes in other diseases. For instance, a study by Gamazon et al. integrated MR and colocalization to identify druggable targets in complex diseases [60]. Huang et al. applied MR in inflammatory diseases, validating their findings via multi-omics data integration [61]. Additionally, research on osteoarthritis and Sjogren syndrome has successfully leveraged multi-omics MR and Bayesian colocalization to pinpoint therapeutic targets [62,63]. These examples validate the robustness of this approach, as it has been successfully replicated across different diseases.

### 3.5. Experimental Validation of Target Genes

The mRNA expression levels of *ORM1*, *MATN2*, and *OVGP1* were quantified in various cell lines via qPCR. The mRNA levels of these genes were significantly lower in RPMI-8226, NCI-H929, OPM-2, and MM.1S cells than in GM12878 cells (*p* < 0.05), suggesting their potential pathogenic roles in MM (Figure 4A–C).

Western blot analysis was performed to validate the mRNA expression trends reflected at the protein level (Figure 4D). The ORM1 and OVGP1 protein levels were significantly lower in MM cell lines than in GM12878 cell lines (*p* < 0.05), indicating the role of these genes in MM pathogenesis and progression (Figure 4E,G). In contrast, MATN2 exhibited an inverse trend; although its mRNA levels were lower in MM cell lines, the WB results revealed significantly higher protein expression in these lines than in GM12878 cells (*p* < 0.05) (Figure 4F). The full-length Western blot images of ORM1, MATN2, OVGP1, and GAPDH are provided in Appendix A, including the molecular weight markers.

ORM1: this has been previously identified as an inflammatory biomarker of MM, but has never been linked to MM progression [64]. OVGP1: this has not been previously associated with MM, but is known as a tumor suppressor in ovarian cancer, where low OVGP1 expression correlates with poor prognosis [65,66].

### 3.6. MR-PheWAS Explored Off-Target Effects of Target Genes

The potential side effects of *ORM1* and *OVGP1* as drug targets were investigated through MR-PheWAS analysis for all 1403 phenotypes in the UKB-SAIGE database (Appendix A). IVW methods with FDR correction revealed significant associations of *ORM1* with phenotypes of blindness, low vision, and acute posthemorrhagic anemia (FDR < 0.05) (Figure 5A). Thus, *OVGP1* was significantly associated with chronic pain and disorders of parathyroid glass (FDR < 0.05) (Figure 5B).

These findings suggest that ORM1 and OVGP1 modulation should be carefully evaluated for potential off-target effects in clinical applications. However, the identified associations do not suggest immediate life-threatening risks, indicating the feasibility of ORM1/OVGP1-based therapies with appropriate monitoring. These results align with previous MR-PheWAS analyses that have explored genetic targets for drug repurposing and adverse event prediction [67].

### 3.7. Pregnanolone and Irinotecan as Effective Agonists of ORM1 and OVGP1

As ORM1 and OVGP1 are negatively correlated with MM progression, potential agonists of these targets were predicted via the DSigDB database. Molecular docking through AutoDock Vina identified pregnanolone and irinotecan as effective agonists for ORM1 and OVGP1, respectively (Appendix A). The docking complexes with the lowest binding energies were visualized via PyMOL (Figure 6).

### 3.8. Effects of Pregnanolone and Irinotecan on Multiple Myeloma Cell Viability

CCK-8 assays demonstrated a significant decrease in the viability of RPMI-8226 and NCI-H929 cells after pregnanolone and irinotecan treatment. The IC50 values for pregnanolone and irinotecan were 32.97 μM and 30.34 μM, and 29.79 μM and 26.97 μM, in RPMI-8226 and NCI-H929 cells, respectively. Subsequently, 30 μM pregnanolone and 25 μM irinotecan were used to treat objective cells (Figure 7A–D). After 24 h, the ORM1 protein levels were significantly greater in the pregnanolone-treated cells than in the control cells (*p* < 0.05) (Figure 7E,F). Similarly, OVGP1 protein levels were significantly greater in the irinotecan-treated cells than in the control cells (*p* < 0.05), indicating that pregnanolone and irinotecan could increase ORM1 and OVGP1 expression, thereby suppressing MM growth (Figure 7G,H). The full-length Western blot images of ORM1 and GAPDH are provided in Appendix A, and the full-length Western blot images of OVGP1 and GAPDH are provided in Appendix A, including molecular weight markers.

Previous studies have explored the use of pregnenolone derivatives in neurological disorders and irinotecan as a topoisomerase inhibitor in cancer treatment [68,69]. However, our study presents the first evidence of these compounds’ potential regulatory effects on ORM1 and OVGP1 in MM.

## 4. Discussion

In this study, we applied a two-sample MR framework to identify and prioritize druggable genes that have a causal impact on multiple myeloma. Starting from thousands of candidate genes, we selected 34 genes associated with MM risk through MR, and then further validated these genes using a TWAS by integrating GWAS summary statistics with gene expression prediction models from GTExv8.EUR whole blood data. The TWAS enhanced the interpretability of the GWAS findings by linking disease-associated variants to gene expression regulation, thereby helping to pinpoint functional target genes. Colocalization analysis (PPH4 > 0.75) highlighted three top gene candidates—*ORM1*, *MATN2*, and *OVGP1*—with strong evidence that genetic variation in these genes drives MM susceptibility. Functional assays confirmed that *ORM1* and *OVGP1* are expressed at significantly lower levels in MM cells than in non-malignant cells, consistent with the notion that their reduced activity may contribute to MM pathogenesis. This finding is in line with the results of the single-tissue TWAS, in which both *ORM1* and *OVGP1* showed significant associations with MM risk, and increased genetically predicted expression was associated with decreased disease risk. We identified pregnenolone and irinotecan as compounds that can upregulate *ORM1* and *OVGP1*, respectively, and showed that these agents inhibit MM cell viability in vitro, while restoring the expression of their target proteins. Together, these findings support ORM1 and OVGP1 as potential therapeutic targets in MM, and suggest a repurposing opportunity for pregnenolone and irinotecan as modulators of these targets. We also leveraged MR-PheWAS to examine the broader effects of *ORM1*/*OVGP1*, and the results did not reveal any prohibitive safety concerns, although some pleiotropic associations were noted, and should be investigated further.

ORM1, also known as α-1-acid glycoprotein, functions as a key protein in the acute-phase response, particularly in inflammation and immune system regulation [70]. ORM1 is produced mainly in the liver, and is released into the bloodstream in response to inflammation, where it serves as an important modulator of immune responses. Its involvement in the tumor microenvironment has been widely recognized, where it influences immune responses through interactions with cytokines and various immune cells [71]. ORM1 has been implicated in both anti-apoptotic and proangiogenic processes across various cancers, suggesting its potential role in tumor progression. Additionally, its interaction with inflammatory cytokines, such as Interleukin 1 Beta (IL-1β) and Interleukin 6 (IL-6), could establish a complex immune network, where ORM1 might either promote or alleviate inflammation, depending on the specific context [72,73]. To the best of our knowledge, our study provides the first genetic evidence linking *ORM1* to MM. Uniquely, we observed the downregulation of *ORM1* in MM cell lines compared to a non-malignant baseline, implying that in MM, *ORM1* might function as a protective factor whose loss promotes disease. This contrasts with reports suggesting that elevated *ORM1* expression correlates with chemotherapy resistance and increased metastasis in certain solid tumors, such as breast cancer and hepatocellular carcinoma, highlighting how the role of a protein can vary across different malignancies [74,75]. The beneficial effect of increasing ORM1 in our MM models (via pregnenolone treatment, which raised ORM1 levels and slowed MM cell growth) highlights a potential therapeutic strategy: enhancing ORM1 activity might suppress MM progression. The exact mechanism by which *ORM1* downregulation favors MM remains to be elucidated. It may involve diminished immunoregulatory support, and since ORM1 can dampen excessive inflammation, its low levels in MM could contribute to a chronic inflammatory milieu that supports tumor growth. ORM1 has been implicated in the PI3K/AKT and ERK pathways [76], which regulate proliferation and apoptosis resistance. Decreased *ORM1* expression may impair negative feedback mechanisms, giving MM cells a survival advantage. Additionally, low levels of ORM1 might impair the negative feedback for angiogenesis or cell survival signals, giving MM cells a growth advantage. These hypotheses merit further investigation, possibly through *ORM1* overexpression or knockdown experiments in MM models.

OVGP1 is a secretory glycoprotein predominantly expressed in the reproductive system, especially in the fallopian tubes, where it is crucial in regulating oocyte fertilization and in the early stages of embryo development. Traditionally, OVGP1 was considered to have a reproduction-specific function, but recent studies have revealed its unexpected involvement in various malignancies, including breast, ovarian, and colorectal cancers [77,78]. Although the role of OVGP1 in the reproductive system is well established, its emerging importance in tumor biology highlights its broader function that extends beyond reproduction. While its role in the reproductive tract is well documented, emerging evidence points to its potential contribution to tumor progression. OVGP1 appears to modulate key processes, such as extracellular matrix (ECM) integrity and cell adhesion, both of which are essential for cancer cell migration and metastasis [79,80]. The glycoprotein may influence tumor cell movement by facilitating interactions between tumor cells and the ECM, thereby promoting invasion into adjacent tissues [81]. Indeed, our discussion of OVGP1’s role in MM is somewhat speculative, given that MM is confined to the bone marrow; however, MM cells interact with the bone marrow ECM and stroma, and a protein like OVGP1 might influence these interactions if expressed or present in that niche. Our genetic data strongly suggest that higher *OVGP1* expression protects against MM (*OVGP1* expression is low in MM, and its low genetic expression increases MM risk). We validated that MM cell lines have markedly reduced OVGP1 levels, aligning with a potential tumor-suppressor role for OVGP1 in myeloma. The identification of irinotecan as an agent that increases *OVGP1* expression in cells is intriguing. It is well documented that irinotecan exerts cytotoxic effects by targeting topoisomerase I, leading to DNA damage and apoptosis [82,83]. However, in our study, irinotecan increased *OVGP1* expression, while simultaneously reducing MM cell viability. This suggests that *OVGP1* upregulation might contribute to the apoptotic response in MM cells, potentially by activating endoplasmic reticulum (ER) stress or interfering with survival pathways. The tumor microenvironment in MM is highly immunosuppressive, allowing MM cells to evade immune detection [84,85]. Some glycoproteins, including mucins, are involved in immune modulation by interacting with immune checkpoint molecules [86,87]. If OVGP1 participates in immune recognition processes, its downregulation in MM could contribute to immune evasion. Future studies should explore whether OVGP1 interacts with immune cells in the bone marrow, such as T cells or macrophages, and whether its expression influences anti-tumor immunity.

Our MR-PheWAS analysis provided additional context for ORM1 and OVGP1 by highlighting phenotypes that might be affected by these proteins. The association of ORM1 with visual impairment and anemia could relate to ORM1’s known function in modulating inflammation; chronic inflammation is an influencing factor in chronic disease conditions like macular degeneration or anemia, and thus, genetically higher levels of *ORM1* (anti-inflammatory) might be protective in these contexts. For OVGP1, the link to chronic pain could be coincidental or via an unknown pathway; the parathyroid association is also not straightforward, though one could speculate about calcium-binding properties of mucins or indirect endocrine effects. These findings are hypothesis-generating, and suggest that the systemic effects of modulating ORM1/OVGP1 should be carefully monitored. Nonetheless, none of the MR-PheWAS hits are acute life-threatening conditions; thus, targeting ORM1 or OVGP1 in MM appears feasible from a safety standpoint, provided that any chronic implications (vision-related, pain-related, metabolic) are managed.

Despite the promising results, our study has several limitations. First, the MR analysis was based on publicly available GWAS summary data, predominantly from individuals of European ancestry. This may limit the generalizability of our findings to other populations with different genetic architectures. Future replication in diverse cohorts or using ancestry-specific GWASs would bolster the evidence for these targets and ensure they are relevant globally. Additionally, while MR is powerful in inferring causality, it can be affected by biases such as residual confounding (if genetic instruments affect the outcome through other traits) and the winner’s curse (the overestimation of effects in discovery datasets). We attempted to mitigate these issues through pleiotropy tests and validation steps, but unmeasured confounders could still have influenced the results. Second, although TWAS and colocalization analyses support the causal roles of *ORM1* and *OVGP1* in MM, these computational methods rely on certain assumptions and the quality of input data. Our experimental validation is still preliminary, although it confirms the differential expression of target genes. We made useful correlations (low *ORM1*/*OVGP1* expression is present in MM cells, and their upregulation reduces MM cell viability), but we have not yet deciphered the mechanism behind the effect of pregnenolone or irinotecan on cell viability that is mediated by *ORM1*/*OVGP1*. Further experiments such as rescue or knockdown studies would be valuable; for example, an assessment of whether silencing *ORM1* inhibits the effect of pregnenolone on MM cells. Such studies would solidify the causal chain from gene to phenotype in a biological sense. Third, the identified compounds (pregnenolone and irinotecan) are existing drugs with known profiles, which is advantageous for repurposing, but their use in MM is novel. Pregnenolone is a precursor steroid hormone, and has been explored mostly for neurological conditions; its effects on immune cells or cancer cells are not well characterized, and could involve off-target pathways. Irinotecan is a chemotherapy agent, and using it at lower doses to modulate *OVGP1* might or might not separate its *OVGP1*-related effects from its DNA-damaging cytotoxicity. Any clinical applications would require careful dosing strategies, and possibly the development of more specific analogues that preferentially enhance OVGP1 without severe toxicity. Fourth, a key consideration in our expression analysis is the choice of GM12878 lymphoblastoid cells as a control group. While GM12878 is widely used in hematologic malignancy studies, it presents several limitations: GM12878 cells are Epstein–Barr virus (EBV)-transformed B lymphocytes, which may not fully reflect the gene expression profile of normal plasma cells. Plasma cells and lymphoblastoid cells have distinct transcriptional landscapes, which could lead to baseline expression differences unrelated to MM pathogenesis. Primary normal plasma cells are difficult to obtain, making GM12878 a practical alternative, albeit with certain caveats. To mitigate these concerns, we validated *ORM1* and *OVGP1* expression across multiple MM cell lines, ensuring the consistency of differential expression patterns. However, future studies should use primary bone marrow-derived plasma cells from healthy donors or integrate single-cell RNA sequencing datasets to establish a more accurate baseline [88,89]. Finally, MM is a complex and heterogeneous disease, and our study focused on single-gene targets in isolation. In reality, MM pathogenesis involves networks of interactions within the bone marrow microenvironment. Proteins like ORM1 and OVGP1 might influence MM progression through multiple mechanisms, for example, by modulating immune evasion, affecting cell adhesion in the marrow niche, or altering sensitivity to other drugs. Fully understanding their roles will require more comprehensive models, potentially co-culture systems with bone marrow stromal cells or in vivo models, to capture the context in which these genes operate. Additionally, future studies should investigate how these targets interact with known MM drivers, and whether combining an ORM1/OVGP1-targeted approach with existing therapies (like proteasome inhibitors or immunotherapies) yields synergistic benefits.

## 5. Conclusions

Multiple genetic analyses and experimental validation were integrated to elucidate the critical roles of ORM1 and OVGP1 in MM. This study also identified two effective agonists, irinotecan and pregnenolone, that target the aforementioned proteins, thereby broadening the scope of therapeutic targets for MM. These results increase our understanding of MM pathogenesis and provide a basis for future drug development. Future research should focus on the specific signaling pathways associated with ORM1 and OVGP1 and their potential in MM treatment, with the aim of developing more precise and effective strategies for the treatment of patients with MM.

## Figures and Tables

**Figure 1 biomedicines-13-00885-f001:**
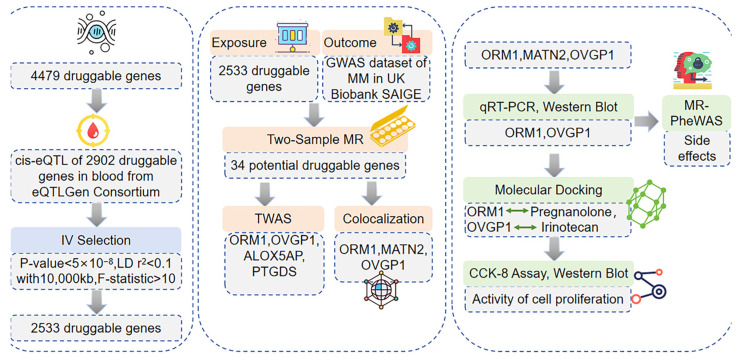
A diagram illustrating the design of our study.

**Figure 2 biomedicines-13-00885-f002:**
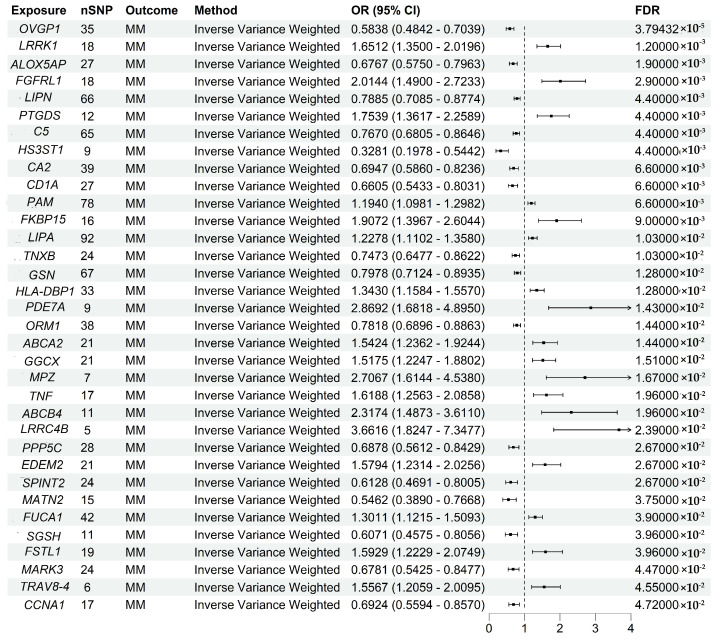
Key Mendelian randomization findings showing the association of druggable gene expression with susceptibility to multiple myeloma, adjusted for false discovery rate correction.

**Figure 3 biomedicines-13-00885-f003:**
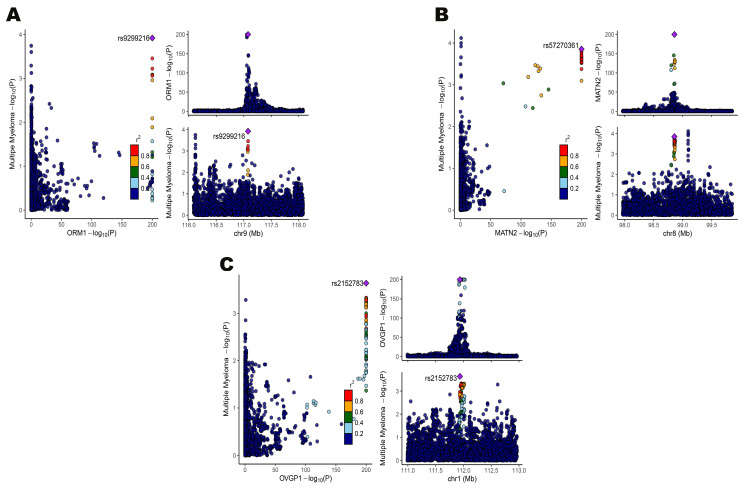
(**A**) Colocalization results of Orosomucoid 1 *(ORM1)* with MM; (**B**) colocalization results of Matrilin-2 *(MATN2)* with MM; and (**C**) colocalization results of Oviductal Glycoprotein 1 *(OVGP1)* with MM.

**Figure 4 biomedicines-13-00885-f004:**
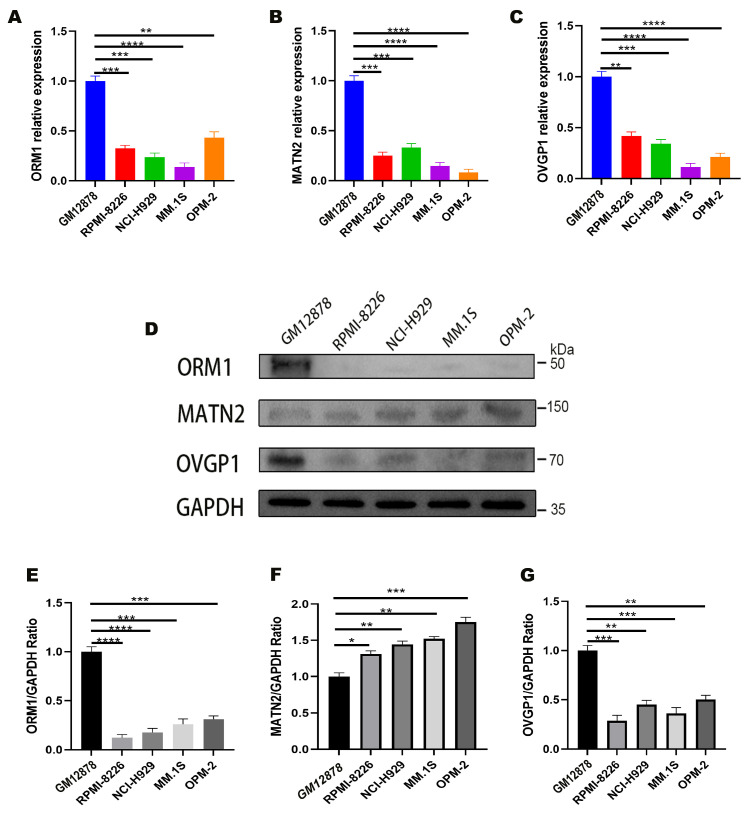
(**A**–**C**) qRT-PCR analysis of *ORM1*, *MATN2*, and *OVGP1* gene expression in GM12878, RPMI-8226, NCI-H929, MM.1S, and OPM-2 cells, with expression levels normalized with GAPDH; (**D**) Western blot analysis of ORM1, MATN2, and OVGP1 from various cells; (**E**–**G**) quantification of ORM1, MATN2, and OVGP1 levels after normalization with GAPDH levels.* indicates *p* < 0.05; ** indicates *p* < 0.01; *** indicates *p* < 0.001; **** indicates *p* < 0.0001.

**Figure 5 biomedicines-13-00885-f005:**
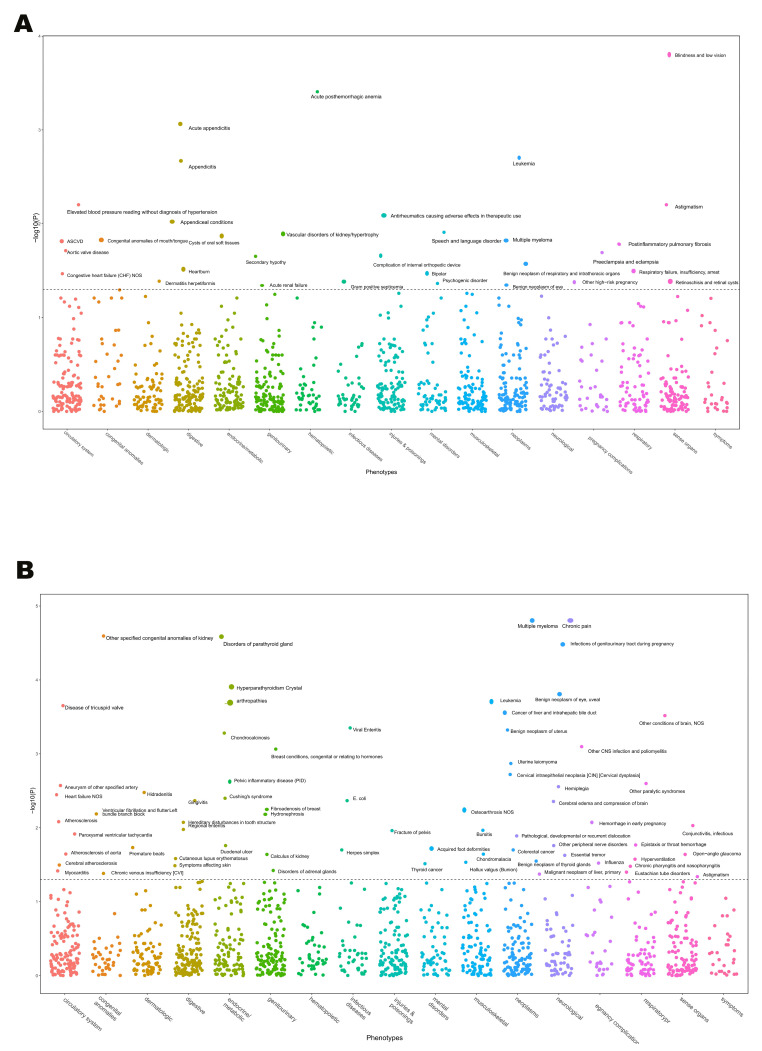
(**A**). MR-PheWAS results for *ORM1* and UKB-SAIGE-associated phenotypes; (**B**) MR-PheWAS results for OVGP1 and UKB-SAIGE-associated phenotypes.

**Figure 6 biomedicines-13-00885-f006:**
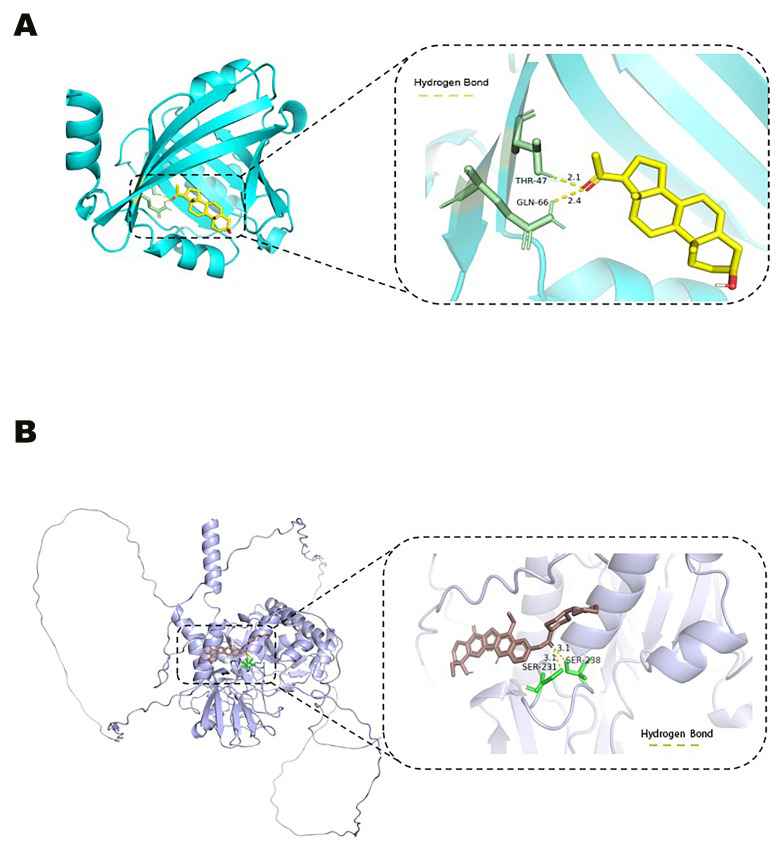
(**A**). Visualization of molecular docking between ORM1 and pregnanolone (hydrogen bonds are represented by yellow dashed lines); (**B**) visualization of molecular docking between OVGP1 and irinotecan (hydrogen bonds are represented by yellow dashed lines).

**Figure 7 biomedicines-13-00885-f007:**
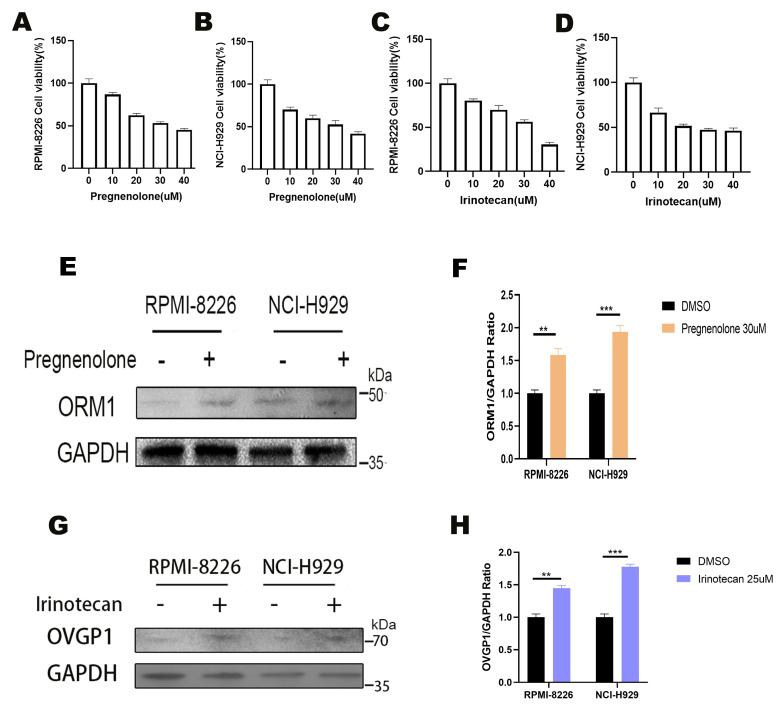
(**A**,**B**). Effects of pregnanolone on viability of RPMI-8226 and NCI-H929 cells; (**C**,**D**) effects of irinotecan on viability of RPMI-8226 and NCI-H929 cells; (**E**) effects of pregnanolone on ORM1 protein expression levels in RPMI-8226 and NCI-H929 cells; (**F**) quantification of ORM1 normalized with GAPDH; (**G**) effects of irinotecan on OVGP1 protein expression levels in RPMI-8226 and NCI-H929 cells; (**H**) quantification of OVGP1 normalized with GAPDH. ** indicates *p* < 0.01; *** indicates *p* < 0.001.

**Table 1 biomedicines-13-00885-t001:** Sequences of primers for ORM1, MATN2, OVGP1, and GAPDH.

Gene	Forward (5′–3′)	Reverse (5′–3′)
*ORM1*	ACCTACATGCTTGCTTTTGACG	CCCCCAAGTCTCTGTCCTGA
*MATN2*	GTGTCAACACCCATGACTATGC	CATCAGGACCAATGTCCAAG
*OVGP1*	AGCGAAGAAGCACTGGATTGA	ATTCACAGCAGATGACAGCCA
*GAPDH*	GAAGGTGAAGGTCGGAGTC	GAAGATGGTGATGGGATTTC

**Table 2 biomedicines-13-00885-t002:** TWAS analysis of MR-prioritized candidate genes.

ID	CHR	SNPID	pos	Z	FDR
*OVGP1*	1	rs1264878	111947430	−3.442623	0.013824
*ORM1*	9	rs7851482	117078286	−2.95	0.03416
*ALOX5AP*	13	rs6490461	30976845	−2.8571	0.03416
*PTGDS*	9	rs2811786	139683224	2.75	0.03594

## Data Availability

All datasets used and analyzed in this study are publicly available. The genetic association data for multiple myeloma (MM) were obtained from the UK Biobank (https://pheweb.org/UKB-SAIGE/, accessed on 28 August 2024), while the cis-eQTL data were retrieved from the eQTLGen Consortium (https://www.eqtlgen.org/, accessed on 7 September 2024). The transcriptomic prediction models for the single-tissue TWAS (GTExv8.EUR.Whole_Blood) were obtained from the FUSION website (http://gusevlab.org/projects/fusion/, accessed on 19 March 2025). The protein function and annotation data were obtained from the UniProt database (https://www.uniprot.org/, accessed on 19 September 2024), and the drug–gene interaction data were sourced from the DSigDB database (https://maayanlab.cloud/DSigDB/, accessed on 6 October 2024).

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
