# Peer review of "Therapeutic Target Discovery for Multiple Myeloma: Identifying Druggable Genes via Mendelian Randomization"

_biomedicines, 2025, doi:10.3390/biomedicines13040885_

Round 1
Reviewer 1 Report
Comments and Suggestions for Authors
In the submitted manuscript, the authors aim to identify novel therapeutic targets for multiple myeloma (MM), addressing the urgent need for new strategies due to relapse and drug resistance. They use Mendelian randomization (MR) and other genetic analysis techniques (SMR, colocalization) to identify potential druggable gene targets for MM. The authors identified nine druggable genes via MR, with ORM1, MATN2, and OVGP1 further validated through colocalization analysis.
Overall, the quality Overall, the quality of the manuscript is good, with the data presented clearly. The data analysis is comprehensive, and importantly, the results are supported by in vitro experimental validation.
A few points should be considered to strengthen the paper:
1. Can the authors provide more insight into the mechanism by which ORM1 and OVGP1 affect MM cell viability?
2. The authors should provide more details about the molecular docking studies and functional assay in the methods section.
3. The authors need to improve the writing further, proof read the manuscript and correct grammatical errors.
Comments on the Quality of English LanguageThe authors need to improve the writing further, proof read the manuscript and correct grammatical errors.
Reviewer 2 Report
Comments and Suggestions for Authors
This study identifies potential therapeutic targets for multiple myeloma using Mendelian Randomization, focusing on druggable genes. Key genes such as ORM1 and OVGP1 were validated through qPCR, Western blotting, and colocalization analysis, providing promising insights for future MM treatments. However, the following aspects need to be revised:
- Data and code need to be shared either through a code-sharing repo like GitHub or a docker-like system such as codeocean for clear reproducibility of the work.
- Author should use STROBE-MR checklist to improve reporting of MR studies and cite PMID: 37198682
- The p letter for statistical analysis should be uppercase - italic face letter. Revise throughout the MS
- In line 223, Two MR-PheWAS studies (PMID: 37046283, PMID: 37013595) are recommended to be cited to expand the content.
- The author is advised to polish the English.
Reviewer 3 Report
Comments and Suggestions for Authors
The manuscript is well-written and discusses good ideas for treating blood cancers, such as Multiple myeloma (MM), a hematological malignancy originating from plasma cells in the bone marrow. Still, it needs to be reviewed and some points improved, including the following:
Please see the report,

Reviewer 4 Report
Comments and Suggestions for Authors
Peer-Review
Therapeutic Target Discovery for Multiple Myeloma: Identifying Druggable Genes via Mendelian Randomization
Title
No comments.
Abstract
No comments.
Introduction
Recommendation : Please expand on the details of Mendelian randomization and drug-target Mendelian randomization in the introduction, as these topics are crucial to the manuscript.
Recommendation : Please include a clear statement of the study’s objective at the end of the introduction.
Methods
Recommendation: Consider revising the section titles to avoid acronyms. For example, instead of using "2.4. IV Selection," "2.6. Two-Sample MR," "2.7. SMR," and "2.9. MR-PheWAS," please write out the full terms. Although these acronyms are defined elsewhere in the manuscript, they are not as widely recognized as terms like RT-qPCR, so using their full names in the section titles will enhance clarity.
Recommendation : Please check if the following sentence at the end of the Methods section is necessary: 'Studies involving interventional procedures with animals or humans, and other research requiring ethical approval, must include the name of the approving authority and the corresponding ethical approval code.'
Recommendation : Please ensure the correct nomenclature is used for RT‑qPCR experiments and adhere to the MIQE guidelines for gene expression analysis by RT‑qPCR (see Bustin et al., 2009 for reference). As recommended by the MIQE guidelines, please include the correct nomenclature (e.g., RT‑qPCR, PCR efficiencies, and other essential parameters) to ensure transparency and reproducibility.
Recommendation: The manuscript lacks a detailed description of the statistical analysis used for RT‑qPCR experiments.
Recommendation: The Methods and Results sections present a very direct approach to describing the analyses and results, yet the underlying analyses are highly complex. It would be beneficial for readers who may be less familiar with the subject if the authors provided a more detailed description of their bioinformatic and genetic screening processes. For instance, additional information on the software, versions, and specific parameters used for filtering cis‑eQTLs and setting inclusion/exclusion criteria—along with a clear justification for the IV thresholds (e.g., P < 5 × 10⁻⁸, r² < 0.1, a 10,000 kb window, and F statistic > 10)—would clarify how these choices contribute to the robustness of the Mendelian randomization analysis. Furthermore, an in-depth explanation of the sensitivity tests (such as Cochran’s Q for heterogeneity and MR‑Egger for pleiotropy), more details on the colocalization analysis methodology (including the assumptions of the “coloc” package and the rationale for the PPH4 > 0.75 threshold), as well as precise information on the parameters used in molecular docking (e.g., grid box settings, scoring functions, and selection criteria for the best complexes) would enhance transparency. A thorough description of the in vitro validation methods—including the number of replicates, statistical tests, and control conditions—would also greatly improve the reproducibility of the study and better prepare the reader for the complexity of the analyses performed.
Results
Recommendation: The manuscript outlines a multi-step pipeline—encompassing the identification of exposure genes (3.1), Two-Sample MR (3.2), SMR (3.3), colocalization (3.4), and experimental validation (3.5)—to identify ORM1, MATN2, and OVGP1 as potential targets. Please clarify whether this integrated approach is novel or based on previously established protocols, and comment on the method’s validity. Specifically, it would be helpful to know if this pipeline (or components of it) has been successfully applied to identify other druggable targets in different contexts, to further underscore the robustness and generalizability of your method.
Discussion
Recommendation: it would be beneficial to include a brief discussion highlighting how these findings compare to or build upon existing literature. For instance, the authors could briefly remark on whether the identified genes or pathways have been noted in prior studies.
Recommendation : Please include in the Discussion a comment addressing the potential impact of using GM12878 cells as controls. Although these cells are a practical and commonly used option, their status as transformed B lymphocytes might not fully capture the characteristics of plasma cells. Could you discuss whether this limitation might have influenced your gene expression results and how it should be considered when interpreting the data?
Recommendation: The MR‑PheWAS results are not discussed in the manuscript. Please include a discussion on these findings, addressing their implications for potential off-target effects and their relevance for validating the identified therapeutic targets.
Conclusion
No comments.
Round 2
Reviewer 2 Report
Comments and Suggestions for Authors
The author basically revised the big problem of the article, and there are some details that need to be improved
1. Summary-based Mendelian Randomization Analysis (SMR) should be replaced with transcriptome-wide association studies (TWAS). The main purpose of TWAS is to identify disease susceptibility genes. (PMID: 37658226)
2. In Figure 1 and Figure 2, "5E-08" should be changed to "5×10⁻⁸".
3. In Figure 2 and Figure 3, gene names should be italicized.
4. Previous researchers have integrated plasma proteomes with genome-wide association data for causal protein identification in multiple myeloma (PMID: 37775746), which should be introduced in the background.
